# Scalable Lightweight IoT-Based Smart Weather Measurement System

**DOI:** 10.3390/s23125569

**Published:** 2023-06-14

**Authors:** Abdullah Albuali, Ramasamy Srinivasagan, Ahmed Aljughaiman, Fatima Alderazi

**Affiliations:** 1Department of Computer Networks and Communications, College of Computer Sciences and Information Technology, King Faisal University, Al-Ahsa 31982, Saudi Arabia; aaaljughaiman@kfu.edu.sa; 2Department of Computer Engineering, College of Computer Sciences and Information Technology, King Faisal University, Al-Ahsa 31982, Saudi Arabia; rsamy@kfu.edu.sa; 3Department of Computer Science, College of Computer Sciences and Information Technology, King Faisal University, Al-Ahsa 31982, Saudi Arabia; 220001672@student.kfu.edu.sa

**Keywords:** low-cost weather station, wind speed, wind velocity, decision tree, meteorology, tiny machine learning (TinyML), conventional neural network

## Abstract

The Internet of Things (IoT) plays a critical role in remotely monitoring a wide variety of different application sectors, including agriculture, building, and energy. The wind turbine energy generator (WTEG) is a real-world application that can take advantage of IoT technologies, such as a low-cost weather station, where human activities can be significantly affected by enhancing the production of clean energy based on the known direction of the wind. Meanwhile, common weather stations are neither affordable nor customizable for specific applications. Moreover, due to weather forecast changes over time and location within the same city, it is not efficient to rely on a limited number of weather stations that may be located far away from a recipient’s location. Therefore, in this paper, we focus on presenting a low-cost weather station that relies on an artificial intelligence (AI) algorithm that can be distributed across a WTEG area with minimal cost. The proposed study measures multiple weather parameters, such as wind direction, wind velocity (WV), temperature, pressure, mean sea level, and relative humidity to provide current measurements to recipients and AI-based forecasts. In addition, the proposed study consists of several heterogeneous nodes and a controller for each station in a target area. The collected data can be transmitted through Bluetooth low energy (BLE). The experimental results reveal that the proposed study matches the standard of the National Meteorological Center (NMC), with a nowcast measurement of 95% accuracy for WV and 92% for wind direction (WD).

## 1. Introduction

In recent years, the rapid change in weather conditions has played a vital role in a number of applications, such as weather perception, agriculture, transportation, logistics, buildings, and energy [1,2]. Nowadays, electricity is an essential service that humans utilize on a daily basis. As the number of electronic devices has increased dramatically over the last decade, it has become a challenge to provide sufficient electricity to recipients with minimal shortage. In 2050, the population of the world is projected to range approximately between 9.4 billion and 10.1 billion [3]. Therefore, it is necessary to meet the global needs for electricity. To meet this need, substantial global efforts are being made to generate clean power based on the WTEG.By 2050, 15–25% of global electricity will be from the WTEG [4]. Further, natural disasters are occurring more frequently globally due to rapid environmental change, such as tornados, hurricanes, waves, storms, and extreme rainfall and snow, all of which are considered unsafe for humans [5,6,7,8]. Therefore, real-time monitoring is essential to reduce the number of deaths due to a catastrophic event [8,9]. In order to avoid a catastrophic event that may damage the economies of countries worldwide, numerical Weather Prediction (NWP) is critical to assess the prevailing weather conditions to make the right decision from the perspective of threat and risk management [1]. Therefore, the accuracy of the predicionsof a weather station is of utmost importance. There are four forecasting techniques used to predict weather: persistence, synoptic, statistical, and computer [5]. Persistence forecasting relies on current weather status to predict tomorrow’s weather status; this is considered the most fundamental method to predict weather.Synoptic forecasting takes into account the principles of measurements, while meteorologists use this information in addition to their perception to predict weather at a specific time. Statistical forecasting involves metereologists taking the normal records of several parameters, such as temperature, wind speed, WD, rainfall, and snow over several years to enable forecasters to predict weather at a specific time. Computer forecasting registers the perception of meteorologists and extreme weather conditions into a computer to predict the weather for the next few days. Commonly, meteorologists utilize computer-based techniques in conjunction with the use of other techniques. The computer-based technique likely results in the most accurate predictions, as it requires a lower amount of intervention by meteorologists compared to other techniques.

Further, weather prediction can be classified into four classes: nowcast, short-term, medium-range, and long-term [7]. Nowcast forecasting refers to the prediction of current weather. Short-term prediction implies weather prediction for the next 72 h. Medium-range prediction refers to the prediction of weather for the next three to seven days. Long-term forecasting implies the prediction of weather fo the upcoming weeks, months, and years.Nowadays, numerous studies rely on machine learning (ML) to address the issues of weather prediction. In ML, the algorithms can be either online or part of a trained model. An online learning model is an algorithm that constantly learns and adapts its learning to upcoming data. In contrast, the trained model is an algorithm that requires a learning phase based on a large amount of data before being able to predict weather.This paper focuses on measuring the nowcast of WV and WD using AI, eight IoT stationary sensors, and a TinyML kit.We implemented our study in multiple stages—dataset, impulse, test, and deploy. In the first stage, we collected the data. In the second stage, we preprocessed the collected data, designed our model, and trained our model. In the third stage, we evaluated and optimized the results of WV and WD. In the last stage, we deployed our model using the Edge Impulse platform.

### Objectives

The core objective of this study is to propose a lightweight IoT-based weather station that relies on AI to measure WV and WD.Thus, we propose the following hypotheses:The proposed system has investigated different scenarios to enable results to benefit multiple applications.The proposed system relies on open-source programs to minimize the cost of the weather station. In addition, the advantage of the open source is that it allows us to customize the source code in such a manner that meets our application requirements. Therefore, we used open source hardware, open source integerated development environment, and Tensorflow Lite framework to optimize the use of system resources.The experiment relies on a homogenous hardware (HW) of IoT sensors and a base station(BS) that perform the heavy work to measure the WD and velocity. The BS can communicate with end users via the BLE or wireless communication in order to transmit the results. The IoT sensors are capable of measuring the wind speed, WD, temperature, velocity, and humidity. In addition, the IoT sensors continuously monitor the targeted area, transmit collected data, measure the weather via BS, and notify the targeted application based on its requirements.The IoT sensors are connected to the BS via wires. The BS relies on BLE to transmit the results to the recipient. Therefore, any IoT sensors can be deployed as long as the BS have the capability of wireless transmission to transmit the results to the intended recipient.The proposed system considers the total cost throughout the design lifecycle. The cost of IoT sensors, BS, and software required to conduct the experiment are the key factors in proposing a low-cost weather station system that can satisfy the requirements of different applications.

The remainder of this paper is organized in the following manner: Section 2 provides a concise background, and Section 3 extensively discusses related work. Subsequently, Section 4 introduces the proposed work, and Section 5 presents the results and facilitates a comprehensive discussion. Finally, Section 6 presents both the conclusion and directions for future work.

## 2. Background

A low-cost weather station plays a pivotal role in enhancing human life with regard to numerous aspects. In the last decade, there have been numerous research contributions toenhancing farming crop.However, different applications have different requirements [1,10]. In addition, a major issue in recent contributions is that the location of the weather station in relation to the targeted area is not considered in the experimental results, which can mislead the decision [11]. Therefore, we propose a low-cost weather station for the WTEG sector. The proposed system has been developed to provide real-time or near-real time data that relies on an AI-based algorithm to make a smart decision at the right time. It is rather critical to receive the results of the prediction in real-time, as the prediction will not be helpful after an incident occurs [12].

### 2.1. Layers

The proposed system utilizes IoT sensors that comprise three layers: perception, transmission, and management layers [10].

#### 2.1.1. Perception Layer

The perception layer is first layer of the proposed system. Each weather station consists of eight interconnected sensors and a BS. This layer is responsible for measuring certain aspects of environmental parameters via the deployed sensors. In particular, these sensors measure the wind speed, wind direction, temperature, velocity, and humidity [10]. The BS collects the collected data from the BS via wires. Then, the BS analyzes the collected data through an AI-based algorithm to notify the intended recipient regarding the real-time status of the targeted area.

#### 2.1.2. Transmission Layer

The transmission layer is second layer of the proposed system. As the name implies, it is responsible for transmitting collected data from one component to another. This layer has two responsibilities—first, to transmit the data collected by the deployed sensors to the BS through the connected wires; second, to transmit collected data from the BS to the intended recipient via BLE. The BLE is an energy-efficient technique while transmitting data at a higher data rate compared to other wireless technologies, as presented in Table 1. This is of significance to meet the minimum requirements of real-time applications, such as WTEG.

#### 2.1.3. Management Layer

The management layer is last layer of the proposed design. The main task of this layer is to provide the intended recipient with a real-time analysis of the targeted area. The management layer relies on an AI-based algorithm that requires several parameters to produce a more accurate result of weather forecasting for a targeted area.

### 2.2. Internet of Things

IoT can be utilized for a wide range of applications, including agriculture, transportation, logistics, buildings, and energy [2]. IoT is capable of continuously monitoring a remote environment and analyzing data. Data analysis can be used to make an optimal decision regarding whether shifting WTEG to a particular direction can generate the maximum power. IoT sensors can interoperate with one another across to monitor an area of interest. These sensors can communicate with one another via wired or wireless communication. To establish communication, IoT sensors can utilize both wired and wireless means. The most prevalent wireless technologies for exchanging collected data—particularly in challenging geographical environments such as deserts, mountains, and seas—include Wi-Fi, BLE, and other wireless technologies. Consequently, a wireless interconnection forms the basis of a network comprising numerous IoT sensors.

IoT sensors are considered the main element for sensing an environment and collecting data; therefore, it can be used to automate decision-making to control certain activities, such as appropriate time of harvesting, pump control, irrigation, etc. However, IoT sensors have limited capabilities as they have limited resources, such as central processing units (CPUs), power, and memory. Therefore, relying on IoT sensors to provide a reliable and efficient system that can satisfy multiple applications simultaneously is rather challenging [14,15,16]. This indicates that each application may require different techniques to satisfy the requirements of the target application. To optimize power consumption, the IoT sensors are responsible for collecting the data and transmit them to the BS, which requires minimum resources. The BS then analyzes the collected data from the IoT sensors and broadcast it to the recipient.

IoT-based solutions for a large number of applications are considered the optimal solution and have attracted many researchers. However, numerous studies focus on smart farming and other applications that have their own unique features [2,10,17]. In additon, there are a few limitations of using IoT sensors—cost, different vendors with different capabilities and performance, lack of technical knowledge, and maintenance. These elements can considered obstacles to deploy the IoT sensors by non-IT specialists. Nevertheless, many researchers continuously propose unique low cost weather station systems that targeted different applications.

Further, weather stations can monitor a targeted area using two techniques. The first technique is by collecting real-time data and transmitting it to a BS. The BS then analyzes the collected data and produces a report. On the other hand, Vat et al. [18] developed a distributed sensor network prediction calculations (DSN-PC) method where sensors can collect real-time data and connect via the Internet and should be able to broadcast a report without the need of a BS. Instead of requiring each node to collect monitored data from its neighbors, an iterative solution to conduct the calculation in a distributed manner. This is achieved by requiring sensors to collect data from their neighbors and apply finite difference schemes. However, this may be suitable when the number of sensors are low, as increasing the number of sensors requires nodes to remain in active mode in order to obtain a complete information of the neighbors. In addition, the result of this paper relies on user datagram protocol (UDP), which makes it suitable for certain types of applications. Moreover, certain applications may require transmission control protocol (TCP) to function and, hence, requires higher memory during its operation.

### 2.3. Artificial Intelligence

AI plays a key role in enhancing different aspects of human life. Numerous studies rely on AI to enhance the accuracy of results while minimizing the cost of a weather station system. Since AI and automatic weather station (AWS) make decisions based on data collected on environmental weather, Rasp et al. [19] provided a data set known as WeatherBench to make easier comparisons between different techniques and yield more accurate results in predicting upcoming weather conditions in the medim term (two days to two weeks). Rasp et al. [19] also specified which evaluation metrics should be used to compare between different algorithms. In [20], the authors presented an agriculture weather station system that relies on IoT and AI to make a decision on crop status. In particular, the system helps to identify any potential of epidemic disease. In additon, due to shortage of water in developed countries, Nawandar et al. [21] presented a study that focuses on optimizing water usage. The system relies on message queuing telemetry transport (MQTT), hypertext transfer protocol (HTTP), and neural networks (NN) to make a smart decision regarding which areas of the farm requires water the most compared to other areas. In [22], the authors focused on developing a smart low-cost weather station system for smart farming with a user-friendly system that can notify farmers about the live status of the farm. Similarly, Ramli et al. [23] proposed an adaptive network mechanism for smart farms while utilizing long range wide area network (LoRa-WAN) and IEEE 802.11ac protocols as the channels of communication. Their network utilizes LoRa-WAN to transmit small data sizes, such as data collected from sensors, while IEEE 802.11 ac is used to transmit large data sizes, such as video and audio. The adaptive mechanism relies on both LoRa-WAN and IEEE 802.11ac protocols to enhance reliability in terms of delay and the amounts of data collected from sensors. Furthermore, authors in [24] investigated the energy consumption of IoT sensors while utilizing IEEE 802.11g, IEEE 802.15.4, and LoRa-WAN protocols as a means of transmission.

In recent years, more proposals have been developed that rely on ML and deep learning (DL) to predict weather [6,25]. These methods include support vector machine (SVM), artificial neural network (ANN), extreme learning machine, regression tree, random forest, and hierarchical mixture of experts [11]. By using DL, the NN has been used as a valid method to predict the load. The ML breaks down into two different models—shallow and deep. The main difference between the shallow and deep models is the number of transformations in an output. A shallow ML model transforms the input one or two times to result in an output, while a deep ML model transforms the input multiple times to result an output. Therefore, deep ML can be used to predict more complicated patterns. A great example of deep ML is the multilayer perception (MLP) model. The MLP is not suitable to predict time-series data while recurrent neural network (RNN) does. Both shallow and deep ML can be utilized to predict data that will occur at a later time. The authors developed an AWS that relies on Raspberry Pi Zero (RPi-0), Wi-Fi communication, and Raspbian operating system (OS). Three sensors were used in this study—MCP9808, BMP180, and DHT 22, and they are all are capable of keeping track of temperature.

### 2.4. Automatic Weather Station System

AWS refers to a meteorological station that collects information on rapid changes in weather to meet the requirements of specific applications. The AWS contains a number of sensors to collect data while connecting to microcontroller(s), such as Arduino or Raspberry Pi. With the rapid development of technologies, such as IoT and AI, it is critical to automate the decision-making with as little human intervention as possible. Ioannou et al. [25] conducted an exhaustive survey of the technologies that can be used to monitor weather—such as IoT, edge computing, DL, and low power wide area network (LPWAN)—to assist in developing an AWS for different applications. In addition, the authors discussed the available topologies in AWS. Similar to the topology of a traditional wireless network, AWS can be arranged in point-to-point, bus, star, and mesh topologies. The data acquisition of AWS can be either offline or online. The main difference between the two is that offline data acquisition does not use any transmission technology to transmit collected data to the recipients, while online data acquisition does. Due to the capability of advanced technologies, online AWS is more commonly used than offline AWS. The transmission of the collected data between different components can be through wired or wireless communications. The results of this study revealed that the developed AWS offers 88% accuracy, with a training time at the cloud and edge server for 195 s and 110 min 32 s, respectively. The IoT sensors require six seconds to execute a task. Based on the given results, the developed AWS may be suitable to monitor rough terrain and rural areas.

## 3. Related Work

Different methods have been applied to address the issue of using IoT-based applications for weather prediction. We can classify the related work papers into the following four categories: ML-based, cloud-based, and IoT embedded weather-based. Each one of these papers has a unique solution to address common problems of weather prediction, but each of them uses different techniques.

### 3.1. Machine Learning-Based Proposals for Weather Prediction

Faid et al. [10] proposed a low cost IoT system that relies on AI algorithms for smart farming while requiring low energy. The proposed system keeps track of temperature, humidity, and pressure to produce real-time forecast for a farm. In addition, the proposed system relies on heterogeneous nodes and BSs. The deployed sensors collect data and transmit it to the BSs. Next, the BS analyzes the data and backs up the results in the cloud via wireless communication. For wireless transmission, the system relies on MQTT, Wi-Fi, and NRF24L01. In comparison to many studies that utilize system on chips (SoCs), this work relies on Raspberry Pi microcomputer, which is capable of performing multiple tasks. In addition, it utilizes the long term and short term model (LSTM) and RNN for analyzing collected data [25]. The results of this study reveal that the proposed design provides high performance in terms of real-time monitoring, temperature, wind speed, and others.

Similarly to L et al. [5], Foudour et al. [9] developed an IoT-based AWS based on multiple linear regression (MLR) and k-nearest neighbors (K-NN) to predict weather for twenty minutes to one hour. The authors considered the following meteorological parameters: wind speed, WD, temperature, humidity, pressure, rainfall, and luminosity. The authors also investigated three prediction algorithms, including adaptive and non-adaptive algorithms, to identify the most appropriate one. The experimental results reveal 6.58% errors by using the best adaptive prediction algorithm, while the worst non-adaptive prediction algorithm results in 13.59% error. This indicates that the developed IoT-based AWS has a high accuracy. Moroever, the period time prediction is too short to notify users and make a smart decision in case of a potential catastrophic event.

Karvelis et al. [7] developed a lightweight IoT-based AWS to predict weather in real time for the next thirty minutes to two hours using a linear regression ML algorithm that utilizes online model. This indicates that the proposed model does not require pre-training to predict weather. The proposed algorithm is known as PortWeather that targets maritime transportation applications to reduce the likelihood of accidents and, therefore, minimizes losses. PortWeather aims to measure WD and WV in order to notify a captain about the weather conditions. PortWeather has been tested using real-time data on static AWSs. A comparison of the results has been conducted based on median absolute error (MAE), root mean square error (RMSE), and mean absolute percentage error (MAPE) to ensure accuracy. The experimental results reveal that PortWeather offers great results in comparison to other models in terms of WD and WV.

Scher et al. [26] used the GEFS reforecast for North Atlantic and Europe (20${}^{\circ }$ W to 50${}^{\circ }$ E, 20${}^{\circ }$ N to 80${}^{\circ }$ N) in its second version. The data collected from December 1984 to 2016, with approximately 10,000 samples, applied convolutional neural network (CNN) to predict the spread and error from the atmospheric field. The data was split into three sets: the years 1990 and 2008 for validation, the period 2010–2016 for testing, and all other years for training. The proposed network outperformed two of the baseline methods results—persistence/local dimension and weather type clustering, both in terms of correlation with forecast spread and in classifying forecast error.

Further, Parashar [27] proposed an Arduino system that consists of eight sensors connected to a display device to predict the weather conditions. Cloud computing was used to upload data on thingspeak, Multiple Linear Regression (MLR) on Jupiter Lab data training, after using Matlab for model analysis. The correlation analysis was performed on three parameters—temperature, humidity, and pressure; the temperature contains an error of 0.5 degree celsius, thereby achieving 99.05 accuracy.

#### Deep Learning-Based Proposals for Weather Prediction

Several recent related works have been reviewed to study the utilization of DL to resolve the issues faced by specific applications. Want et al. [11] investigated 12 data-driven models to compare between shallow ML and DL for building thermal load. The authors compared different models in terms of cooling demand. Their study found that XGBoost and LSTM result in higher accuracy in shallow and DL models compared to the baseline model. This indicates that we can use ML techniques to make a wise decision. The baseline model predicts building the thermal load based on a preceding day. However, LSTM is suitable for predicting short time (e.g., one hour ahead). Due to rapid changes in weather, XGBoost results in lower accuracy than LSTM. In contrast, XGBoost offers better results in predicting long-term (e.g., 24 h ahead) compared to LSTM.

Alongi et al. [28], explained how a deep but tiny neural network (DTNN) was created and utilized to make predictions about air pressure. The DTNN is created using a well-known deep learning framework, and these networks are then automatically transformed into a useful C library. This study also shows that the performance estimates made during the design and training phases of the DTNN are consistent with those measured during deployment in a real-world setting. The system was implemented in a real environment for 30 days, from 1 March to 31 March, to assess the system’s performance and integrity. Further, the NRMSE and NMAE values were 0.0328 and 0.0251, respectively.

In Fu et al. [29], the “OBS” and “RMAPS” data sets provided by the committee of AI Challenger in 2018 were used, consisting of data from Beijing weather stations for over three years. The “RMAPS” data set contains 29 meteorological features predicted by the NWP model, while the “OBS” data set contains 9 meteorological features observed by the 10 weather stations. The authors proposed a hybrid model incorporating a (one-dimensional CNN (1D-CNN) and a bidirectional-LSTM (Bi-LSTM) model. It outperformed other experimental algorithms, such as (feedforward neural networks (FNN), 1D-CNN, LSTM, and Bi-LSTM, with a performance of 0.45 RMSE.

Carvajal et al. [30] utilized dengue incidence and meteorological data that were collected in the period from 1 January 2009–31 December 2013 to analyze two types of data sets: observed meteorological factors (MF) and its corresponding delayed or Lagged effect (LG). Four modeling techniques were experimented with—random forest, gradient boosting, general additive modeling, and seasonal autoregressive integrated moving average with exogenous variables. Random forest with delayed meteorological effects (RF-LG) outperformed the other experimented models.

Zhu et al. [31] applied the predictive deep convolutional neural network (PDCNN) for wind speed prediction, which integrates MLP with CNN. The model has been experimented with on the Wind Integration National Dataset (WIND) and the results were compared with the baseline algorithms: CNN, support vector regressor (SVR), decision tree (DT), and persistence method (PR). The results demonstrated the effectiveness of PDCNN, with an RMSE of 0.977.

### 3.2. Cloud-Based Proposals for Weather Prediction

Taso et al. [32] proposed a monitoring AWS system that utilizes IoT for the purpose of students’ learning at Wu-TSO elementary school in Taiwan. This system uses MQTT to transmit monitored data between sensors to the intended recipient instead of storing collected data into a relational database management system (RDBMS) and then visualized on a web server. The sensors are responsible for monitoring an environmental area by collecting data and broadcast the transmitted data to the intended recipient. The proposed system separates the complexity and load on sensors (i.e., data collection and aggregation), database storage, and visualized data viewer. To monitor all aspects of an environment, many sensors must be depolyed, as each one is responsible for monitoring one parameter, such as temperature, wind speed, WD, humidity, rain, and other parameters. Therefore, the authors focused only on four parameters, including WD, wind speed, temperature, and humidity. Therefore, the authors focused only on four parameters, including wind direction, wind speed, temperature, and humidity. The prototype relies on Arduino MKR1000 and Raspberry Pi. Raspberry Pi has been used to create the website and database server. The collected data can be visualized on a web page where end users can analyze the results of the collected data via PHP script in the back end. The main issue of this proposal is that end users cannot monitor weather in real time, as results are updated every 30 min. In addition, there was no comparison made between the predictions of the metereological weather station and their results to ensure the accuracy of the latter.

Similarly, Bin Shahadat et al. [2] developed an efficient IoT-based weather station that utilizes NodeMCU and Blynk IoT technology. This study investigates several parameters of meteorology, such as temperature, pressure, humidity, and rainfall. This low-cost weather station provides end users with knowledge on the weather through the Internet. The proposed system consists of three sensors to predict weather conditions—BMP180 pressure sensor, DHT11 temperature and humidity sensor, and SEN-00194 rain sensor. The system also uses ESP8266 as the Wi-Fi module, cloud, and Blynk-IoT. The sensors collect real data and transmit it to the private cloud over Wi-Fi to the Blynk-IoT for further processing. The experimental result reveals that the proposed weather station offers a great result in terms of accuracy and reliability in comparison to national metereological information.

Math and Dharwadkar [17] proposed an IoT-based automatic weather station (AWS) for precision agriculture (PA) in India. The system monitors real-time data, notifies farmers, and helps optimize resource usage based on crop status. The setup uses low-cost hardware, a microcontroller for data transmission, Arduino for weather station setup, and ThingSpeak for data analysis and visualization. The system includes data collection, storage, and analysis components. ThingSpeak predicts weather conditions, but accuracy and reliability compared to national meteorological data are not evaluated [12].

Similar to [17], Kapoor and Barbhuiya [12] proposed another low cost IoT-based AWS that analyzes the collected data in the cloud. This study focuses on predicting the following parameters: temperature, WV, humidity, pressure, and rainfall. The authors rely on Raspberry Pi Zero for sensors and Raspberry Pi 3 as a BS. The collected data is stored in a database server. The BS transmits collected data and the information available in the database server to the cloud via Wi-Fi. The cloud utilizes ML tools and algorithms to conduct the prediction. By using multiple of sensors and BSs, the proposed system can offer high accuracy, availability, scalability, reliability, portability, and modularity.

### 3.3. IoT Embedded Weather-Based Proposals for Weather Prediction

Ghaderi et al. [4] presented a DL algorithm that relies on RNN to predict spatio-temporal wind speed. A spatio-temporal algorithm relies on collecting information from neighboring sensors to predict weather in a target area. Most researchers do not consider the spatio-temporal issue and, hence, the result may mislead the recipients due to the distance between the monitored area and the location of the targeted application [12]. The authors investigated wind speed to meet the requirements of renewable energy. The contribution of this work is the ability to monitor different nodes over a distributed area using the same framework. Another contribution is the ability to output the result of all nodes simultaneously. The result of this paper proves that the proposed DL algorithm offers a better result for the short-term forecast in comparison to several models through mean absolute error (MAE), root mean squared error (RMSE), and normalized root mean squared error (NRMSE). Thus, DL algorithms can address the shortcomings of traditional methods used nowadays to monitor a targeted area. The authors focused on wind speed to maximize the renewable energy, but this is not enough, as we need to have knowledge of other parameters as well, such as WD.

EL Hachimi et al. [33] proposed a weather data management system that collected real-time data using IoT sensors and stored the collected data in the distributed database MongoDB. They applied the extreme gradient boost (XGBoost) ML model for predicting climate parameters and then tested it with Morocco meteorological data for validation. The model results in forecasting the meteorological data for the year 2020, given the historical data for the period 2013–2019 andt demonstrates a performance of R2 = 0.96 and RMSE = 0.39.

In the above related work, for weather parameter measurements using a rotating anemometer, tipping bucket rain gauge, and ultrasonic-based sensors were used and all of them are costly and require regular maintenance and calibration. Due to the high cost of these sensors, they are only used in a certain locations (meteorological centers, airports, and selected places) which have spatial variability and will not accurately report the weather conditions that are relevant to the concerned locations. In this study, we proposed the creation of TinyML-assisted static sensors using microphone and piezo transducers, which will enable a low cost, low power, and easily scalable weather measurement (WD and WV) deployed at locations of interest, at a distance of 5 km or so to eliminate spatial variability in measuring weather parameters. BLE v5 is used as IoT communication medium in order to save power.

## 4. Proposed Work

The goal of this study is to create a low-cost, low-power, reliable, accurate, and easy-to-install smart weather station, with no mechanical moving parts, for measuring all weather conditions. Our focus is to measure wind parameters based on an ultra-low power ML at the edge. In particular, two wind parameters are utilized in this study—WD and WV.

This study relies on data collected from the AWS of Al-Ahsa International Airport (AIA) in Al-Ahsa, Saudi Arabia [34]. The data set includes several environmental parameters, such as minimum, maximum, and average temperature, wind speed, WD, etc. In order to obtain with most accurate result, this study relies on DT algorithm for WD measurement, which is a fundamental ML concept that can be utilized for classification and regression problems. This technique requires a low training time, has small memory requirements, and has the fastest inference, thereby resulting in high accuracy. The 1D-CNN model is used to measure WV based on the Beaufort wind force scale (BWFS).

The collected data sets cover daily measurements from 1 December 2022 to 31 January 2023. We aim to match our results with the BWFS. The BWFS is an empirical measure that relates wind speed to observed conditions at sea or on land. Table 2 presents the BWFS with speeds in knots, miles per hour, and kilometers per hour. These are mean speeds, usually averaged over 10 min by convention.

### 4.1. System Implementation

This sudy proposes the productio of low-cost, quickly deployable anemometers for determining the direction and velocity of the wind. The conventional-type anemometers typically have moving parts, which increase the maintenance requirements and lead to a poor accuracy over time due to aging and wear and tear. The proposed system pertains to static anemometers without moving parts, whereby such maintenance issues are significantly reduced, thereby minimizing the expense while improving accuracy. Edge Impulse is a powerful TinyML cloud platform that has been utilized to train the proposed model [36].

### 4.2. Hardware Setup of Wind Measurement Sensors

The major requirement of a low-cost weather station is the ability to monitor a targeted area based on predefined parameters. This implies that the weather station might be deployed on normal and rough terrain areas. Therefore, IoT sensors must be able to monitor the targeted area irrespective of its location. The AWS can measure WD, WV, temperature, pressure, mean sea level, and relative humidity. However, this paper focused only on measuring the WV and WD. The sensors were connected to the microcontroller through wires. The microcontroller can further analyze collected data and output the result to the intended recipient via wireless transmission, such as Wi-Fi or Bluetooth. To maximize the lifetime of the proposed weather station, we selected the following hardware components: piezo electric sensor and Arduino Nano 33 BLE sense TinyML kit.

We selected this sensor for WTEG application due to its small size, low cost, and low power requirements. In order to measure the WD, we designed a 3-Dimensional (3D print setup of an anemometer that houses a a set of eight piezo electric pressure sensors (N, NE, E, SE, S, SW, W, NW). The time taken for printing was 11 h. The prototype was fixed at a height of two meters on the roof during experimentation. Each of the piezo sensors have a disk of diameter 12 mm and is mounted on a post in a circular fashion. The diameter of the circular posts is 30 cm, whereas the height of each post facing opposite directions is 10 cm and 8 cm, respectively, thereby avoiding obstruction of the wind flow in different WDs. The width is 1 cm, and the thickness is 0.4 cm.The topview of the 3D printed prototype is presented in Figure 1.

The piezo sensors from all eight directions (N, NE, E, SE, S, SW, W, NW) are connected to A0–A7 of the analog pins of the microcontroller, respectively.To measure the wind velocity, we exploit the omnidirectional microphone embedded in the Arduino Nano 33 BLE sense TinyML kit. The 3D printed prototype was fixed at a height of two meters on the roof during the experimentation.

### 4.3. Technical Details of the Sensors Used for Wind Measurement

The piezo condenser electric transducers are made of brass and ceramic, which sense the force of the wind. The following are the physical dimensions of these transducers: diameter of 12 mm disc, thickness of 0.3 mm, weight of 1 g. The resonant frequency of the sensors are 3.0∼5.0 +/− 0.5 KHz, and the resonant impedance is 300 ohms.

The onboard microphone (MP34DT05-A) in Nano 33 BLE sense is a miniscule, ultra-low power, omnidirectional, Micro Electro Mechanical Structure (MEMS) microphone built with a capacitive sensing element and an Inter-integrated Circuit (I2C). It provides Pulse Density Modulation (PDM) for the sound signal given to the microphone. The Signal to Noise Ratio (SNR) is 64 dB with −26 dBFS ± 3 dB sensitivity. It operates with 3.3v onboard power supply.

We used Arduino Nano 33 BLE Sense-TinyML kit for sensing the direction of winds from piezo transducers and for measuring the real-time inference of WV. Nano 33 BLE is a 32-bit ARM^®^ Cortex™-M4 CPU running at 64 MHz. The main feature of this board, apart from the impressive selection of sensors, is that it supports running edge Computing applications (AI) on it. In addition, the Nano33 BLE sense board hosts a variety of onboard sensors that enable a wide range of TinyML applications.

### 4.4. Methods

Data were collected from 1 December 2022 to 31 January 2023 in different locations of Al-Ahsa in Saudi Arabia (Airport: 25.298534571856415, 49.49630667259468, KFU: 25.340825526993818, 49.5999109794487, Lulu Market: 25.341038358102704, 49.5463214506130 64) at a Mean Sea Level (MSL) of 121 m. The collected data was uploaded into the Edge Impulse cloud training platform for further processing, model development, and testing. Next, the developed model was deployed into the edge device ( TinyML kit) for real-time inference, as shown in the proposed architecture in Figure 2.

The WV and WD measurement was performed using the sound-based DT one-dimensional convolutional neural network (SDT1DCNN) algorithm, as presented in Algorithm 1.

**Algorithm 1:** SDT1DCNN Algorithm.

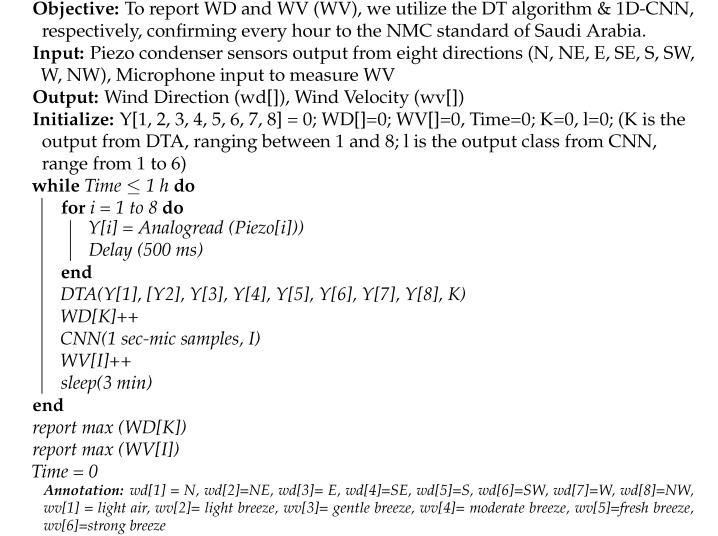



The decision tree (DT) ML algorithm for WD measurement, as shown in Figure 3 is passed with data acquired from all eight piezo sensors to categorize the detected WD on a time slot, as mentioned in our proposed algorithm—the SDT1DCNN. It begins by identifying the sensor that detected the highest WV and then checking for the second highest to report the WD for specific time slots.

The leaf node of the DT terminated with answer “No” signifies that the direction of wind is multi directional and DT is unable to measure the WD. The leaf node terminating with answer “Yes” obtains a conclusive result in that time slot. In every hour, 20 measurements are made by DT and the WD counter is incremented according to the direction of the wind. At the end of reporting time slot (one hour), the direction counter that is at the maximum among all eight directions is reported.

A one-second sound signal of the wind is acquired using a microphone and passed to a 1D-CNN real-time inference engine deployed on the TinyML kit to classify the WV as per BWFS. In every hour, 20 measurements are made by 1D-CNN and the WV counter is incremented according to the six classes of BWFS. At the end of reporting time slot, the WV counter, which is maximum among all six classes ranging from classes 1–6, is reported. The training of the CNN classification model was performed on the Edge Impulse cloud platform [36].

## 5. Results and Discussion

In this section, we present the results of the proposed system in the field. The project experimentation was conducted for the last three months, from December 2022 to February 2023. As previously discussed in our system implementation, two ML algorithms are used. The DT algorithm is used to measure the WD and the 1D-CNN is used to measure the velocity of the wind. We have tested the proposed system using different scenarios (wind velocities) and different locations to ensure its accuracy compared to the meteorological weather station in AIA.

The range of wind velocities during the entire year (from NMC data) varies from 1 mph to 32 mph. Thus, we decided to build a classification model with six classes (C1–C6), as highlighted in the Table 2 of BWFS. In addition, time series samples of one-second duration recorded with microphone of TinyML kit at sampling frequency of 16 kHz during the experimentation phase (unlabeled data) is depicted in Figure 4.

In order to train our classification model for WV as per BWFS ranging from light air (C1) to strong breeze (C6), it is necessary to label the data set respectively from C1 to C6. To label our data set, we first used a digital handheld anemometer to measure the velocity of the wind while collecting the samples for training purposes. The accuracy of the digital anemometer used is +/−5% and can measure the WV up to 65 mph. Next, we also use the data explorer option on the Edge Impulse platform to fine tune the labeling. The data explorer tool helps to label the data before passing the training example to the model. In addition, the spectrogram is a preprocessing block used for extracting the features that help in labeling the data.

### 5.1. Data Preprocessing

The sample data is a sound signal of one-second duration, which is a time series data. This time-series data is passed to the spectrogram block for feature extraction. The spectrogram processing block extracts the time and frequency features of signals. It provides a good performance on any sensor data with continuous frequencies, such as audio data for non-voice recognition.

### 5.2. Spectrogram Parameters

The parameters of the spectrogram are frame length, frame stride, fast fourier transform (FFT) size, and noise floor. The frame length is 0.02 s, the frame stride is 0.01 s, the FFT size is 128, and noise floor is −52 dB. The signal level below −52 dB is considered noise. The spectrogram works in the following manner: It first divides the window in multiple overlapping frames while meeting the above minimum requirements. In our case, for a sample window of one second, the frame length of 0.02 s and stride of 0.01 s will create 99 time frames. With a 16 KHz sampling frequency and a time frame of 0.02 s, each time frame includes 320 samples (16k * 0.02). Since FFT size is 128, the time frames will be truncated to 128 samples. Figure 5 lists the time series and spectrogram of samples for class 1 as per the BWFS.

Using the spectrogram digital signal processing (DSP) block features and WV from the handheld anemometer in mph, the data set utilized for experimentation is labeled and presented in Figure 6.

### 5.3. TinyML Model Development

Building the model started with the default settings and built-in NN architecture in Edge Impulse. The settings have been tuned and retrained to obtain an optimal model fitting for the data in order to achieve the highest accuracy. Table 3 lists the architecture and model parmeters of the NN. We utilize the classification model provided by the Edge Impulse using the keras library and tensor flow framework. In particular, we rely on the 1DCNN layers and fully connected dense layers for the architecture, as depicted in Figure 7. The batch size of 32 is selected in order to reduce memory utilization during the training and inference phases. The learning rate of 0.005 is selected based on batch size and for better convergence. Further, the adaptive momentum (AdaM) optimizer is used because it provides fast convergence.

The Edge Impulse cloud platform is designed to deploy models for real-time applications on the edge device (Arduino Nano 33 BLE). Based on the hyper parameter used for NN model, the inference time is 13 ms and the peak RAM usage is 13.1k out of 256 KB to store model parameters. In addition, the neural network model consumes only 29.4 KB flash usage out of the available 1 MB flash. This number indicates that this model is optimized for TinyML implementation and real-time inferencing.

### 5.4. Evaluation Metrics

The Overall Accuracy (OA) is the fundamental classification performance metric is, for the given data set and it is calculated as follows:(1)$$\mathrm{OA}=\frac{\mathrm{Number}\;\mathrm{of}\;\mathrm{correct}\;\mathrm{measurements}}{\mathrm{Ground}\;\mathrm{Truth}\;(\mathrm{True}\;\mathrm{Value})}$$

However, OA is an important efficiency metric for any classifier, as it presents limited information about the stability and sensitivity of the proposed classifier. Hence, we follow the F-1 score to measure the performance of the models along with OA. The F-1 score is defined as the harmonic mean of the precision and recall of the model. The calculation formula is as given below:(2)$$F1-\mathrm{Score}=\frac{2*\mathrm{Precision}*\mathrm{Recall}}{\mathrm{Precision}+\mathrm{Recall}}$$
(3)$$\mathrm{Precision}=\frac{\mathrm{TP}}{\mathrm{TP}+\mathrm{FP}}$$
(4)$$\mathrm{Recall}=\frac{\mathrm{TP}}{\mathrm{TP}+\mathrm{FN}},$$
where True Positive (TP) = number of true positives, False Positive (FP) = number of false positives, False Negative (FN) = number of false negatives

### 5.5. Experimental Setup

Prior to applying filtering for noise reduction, the accuracy, loss, confusion matrix, and data explorer window reported by the Edge Impulse platform are depicted in Figure 8. The results presented here are for the full training data set. Figure 8a presents the confusion matrix before filtering the surrounding noise, where the accuracy and F1 score are depicted. In Figure 8b, the green blobs represent correct classification while the red blobs represents incorrect ones. The accuracy is only 82.6%. The reason for these accuracy numbers is that the data set was collected at different locations at which surrounding noises exist. Next, we attempted to enhance the accuracy of our classifier by eliminating the surrounding noise. In order to do this, we recorded multiple samples one with a clean environment, while the other one consists of a few noises. Then, the frequency of noices are ascertained and plotted. Figure 9 presents the spectrum plot for the samples collected from the King Faisal University campus when the WV is considered as a light breeze (C2-BWFS) (4–8 mph). We time-stamped the surrounding noise, including cars crossing at high speeds and normal speeds. We also used Matlab to plot time-frequency-spectrum power for the acquired data set. In Figure 10, the x1-axis is time (sample axis), the x2-axis is frequency, and the y-axis (vertical) represents the spectrum power. From this plot, we ascertained that the frequency of the sound signal related to the wind is concentrated at less than 200 Hz, but the frequency of the sound signal related to the cars crossing is approximately 300–600 Hz.Moreover, we identified that human speech and bird noises range between 300 Hz–1.2 KHz. To increase the accuracy of our model, we set the cut off frequency as 300 Hz in the DSP block of thee spectrogram before training the model.

### 5.6. Experiment Results

The accuracy and confusion matrix reported by Edge Impulse for training performance on validation set after applying filtering for noise reduction are depicted in Figure 11. The results depicted here are for the full training set. The diagonal element highlighted in green refers to the individual accuracy for the respective classes. The accuracy has improved by 5.1% compared to that without filtering.

The overall accuracy and F1 score for all the classes as per BWFS are populated in Table 4. The overall accuracy for the trained model is around 93—that is, 558 samples are correctly classified from among 600 samples taken for model testing. To the best of our knowledge, this paper is the first to utilize the microphone and Tiny ML to classify the velocity of the wind. In literature, most researchers rely on ultrasonic rotated anemometer and piezo transducers for the classification of the wind parameter, which reported an accuracy of approximately 90–95. Our proposed technique is also around that band. In addition, the proposed technique ensures low cost, low power, and easily scalable and deployable because of the low-form factor. Figure 12 shows the screenshot of live classification results after the model is deployed on the Nano 33 BLE sense board. A one-second microphone sample was used for live Measurement. The velocity of the wind when performing live classification was 16 mph, which belongs to the C4 class as per BWFS. Figure 12a presents the acquired time series sound signal. The data explorer window featuring the live classification is presented in Figure 12b. In the feature window, the sample acquired is indicated by the light blue blob and classified correctly as C4. Figure 12c summarizes the classification of the live sample. The live sample is classified as C4 (The confidence level for C4 classification is 0.92, the probability of being misclassified as C3 is 0.04, and the probability of being misclassified as C5 is 0.04).

The WV test is conducted with the proposed prototype and compared with NMC data for selected 3 days as shown in Figure 13. Due to restrictions at NMC, we could not place our prototype at the NMC location for fair comparison. However, the results are matching for two days the 9th and 17th of January 2023, but for 31st of January between 10:00 a.m. to 1:00 p.m. there is deviation and that is due to spatial problem.

The results of the prototype for the WD measurement, when tested with a fan in laboratory setup using different directions by shifting the fan to multiple locations are presented in Figure 14. Based on the lab setup, the piezo transducer output from different WD are depicted in Figure 15. These piezo transducer signals are passed to the DT algorithm and the WD results from that are commendable.

Based on the lab setup, the piezo transducer output from different wind sensors versus time are presented in Figure 16. The prototype was first tested in a lab set up, as depicted in Figure 14. The fan was positioned in each direction at a distance of 15 cm from the piezo sensor mounted on the post of the 3D printed WD sensors—North (N), North East (NE), East (E), South East (SE), South (S), South West (SW), West (W), and North West (NW). At each position, 500 ms of data was recorded. The cycle time was four seconds (8 directions * 500 ms).

In Figure 15, Samples 1–20 corresponded to the time duration when the fan was oriented toward N direction, and similarly the next 20 samples for NE, and so on. The four seconds of Piezo transducer signals and the direction of wind were calculated from the DT algorithm. From the wave form, one can visualize that the voltage from piezo sensors is directly proportional to the force of wind incident to it. To ensure the accuracy of our lab setup results, we deployed our prototype on the field. We tested our model for three days in the field on 9, 7, 31 January 2023 from 6:00 a.m. to 6:00 p.m. The results were compared with NMC. We found that on all three days, the results of the WD from our model matched with the NMC results, when the wind velocity was less than 13 mph. On the other hand, the results did not match with the NMC results when the wind speed was greater than 16 mph. One of the reasons for the inaccurate wind direction is that the distance between the piezo sensor was 24 cm. The nearby piezo sensors are highly sensitive and the DT algorithm was unable to infer the correct direction in the sampling time of 500 ms. To enhance the results, a large space is required between the piezo sensors facing opposite directions.

In Figure 16, the plot shows the accuracy and loss curve for both training and validation for 100 epochs during a training cycle. After 90 epochs, the model converges very well and is a good fit. This plot is generated with a seed of 31 (the initial weights and bias during training are random and vary with different seed) during the training stage for consistency in reporting model accuracy.

### 5.7. Discussion

In this section, we present the performance of the proposed system in the field. We tested the proposed system using different scenarios and different locations to ensure its accuracy compared to the metrological weather station in AIA. Different scenarios refer to the data collected over four months (December to March), ranging from wind speeds of 1 Kmph to 45 Kmph, matching classes 1–6 of BWFS, as given in Table 2. In addition, the WV measurement model was developed and tested only for the classes c1–c6 due to the fact that throughout the year the eastern region did not experience gales and violent storms. The designed model produced an overall accuracy of 95%. This accuracy matches the meteorological standards of Saudi Arabia.

Further, the developed prototype can work continuously for 24 h. The TinyML device used in the prototype is Arduino Nano 33 BLE. It runs on 3.3 voltage power supply and contains seven onboard sensors and nRF52840 Bluetooth module (BT version 5). In order to extend the battery life, we switched off all the other sensors except the microphone. The Bluetooth module can transmit data at 2 Mbps, 1 Mbps, 500 kbps, and 125 kbps. After the inference, the result is transmitted to the standard Bluetooth application of a mobile phone at the speed of 125 kbps. We used eight channels of onboard analog-to-digital converter to measure WD. This measurement is only a burst measurement. In an hour, the ADC works only for 80 s (8 (number of directions) * 500 ms (measurement time) * 20 (readings per hour)). During inference, the TinyML kit consumes 120 microamperes, and during transmission 150 microamperes. With this setting for an hour of measurement as per the NMC standard, the designed IoT sensor for WV and WD measurement consumes 3 milliampere hours (mAH). Thus, in a day it consumes 72 mAH. Approximately 100 mAH is sufficient for a single day’s measurement. We used a 3.7-volt, 2400-mAh lithium polimer battery. This battery can support 10 days of autonomy before the system is degraded. This prototype was placed in multiple locations in the Al-Ahsa region for collecting data on the targeted area without the need for battery replacement.

In the case of WD measurement using our 3D printed prototype, we faced accuracy issues in predicting the WD when the WV was greater than 35 Kmph. This is due to the short circumference spacing of piezo sensors. This issue can be resolved by simply increasing the circumference spacing between piezo sensors; however, the prototype becomes bulkier.

The conventional AWS that exists in NMCs is costly and needs maintenance with operators to measure weather parameters. In contrast, the IoT sensors developed in this study are purely static and calibration-free and maintenance-free as well. The cost of the prototype developed in this study is approximately $80. In addition, the lifetim.e of the conventional anemometer is approximately five years, while our sudy relies on microphone-based sensors that have an extended lifetime. In future, we plan to use the temperature, humidity, and barometric pressure sensors of the TinyML kit to assist in measuring weather variations along with developed IoT sensors. Due to the low cost, low power, small form factor, and high accuracy of the measurement system, it is easily scalable and deployable to end users, thereby meeting the objectives.

The result of collected data can assist many applications to make the ideal decision on account of the following aspects:

Reduce the operation expenditure while producing the maximum power in WTEG sector by rotating the WTEG to the right direction.Reduce crop waste by monitoring environmental monitoring parameters, such as temperature, humidity, WD, and WV. This helps to make more profits.Minimize the risks that may face humans during natural disasters by sending an alarm to evacuate a target area.

To summarize, many applications need more accurate results in order to make the right decision; therefore, it is necessary to deploy distributed sensors across a targeted area. Deploying multiple sensors within a targeted area must be affordable and easy to manage and also offer high accuracy. Thus, to benefit most applications, we must study the metrological parameters that affect a targeted application. Apart from the proposed system providing a more accurate result, it minimizes the human interaction required to make a decision. This indicates that our proposed system can satisfy the requirements of real-time application to make real-time decisions.

## 6. Conclusions and Future Work

This paper developed a new model to measure WD and WV using a low cost AWS that can satisfy multiple applications based on a ML algorithm. Due to catastrophic events occuring all over the globe, it is necessary to measure the rapid changes in weather to benefit numerous applications. Some of the existing issues in applications in agriculture, building, and energy can be resolved when real-time meteorological data is available. Following the rapid development of open source software, we used Arduino to conduct this study. The AWS was deployed in multiple locations of Al-Ahsa region to collect real-time data on meteorological parameters. By analyzing the collected data, we designed and developed the proposed algorithm to measure weather parameters. The experimental results reeval that the proposed algorithm achieves more accurate results than the one broadcasted by the NMC. Therefore, the proposed algorithm can assist in making appropriate decisions at the right time.

In our future research, we first plan to investigate our model with other meteorological parameters, such as temperature, pressure, mean sea level, and humidity. Next, we plan to investigate the power consumption of our algorithm compared to others. Then, we plan to analyze mobile AWS to collect real-time data from multiple sources and then decide the averaged weather of a targeted area. Next, we need to create our own model to predict meteorological parameters using stationary AWS and mobile AWS with regression models. Finally, we plan to implement an adaptive auto regression model for noise elimination.

## Figures and Tables

**Figure 1 sensors-23-05569-f001:**
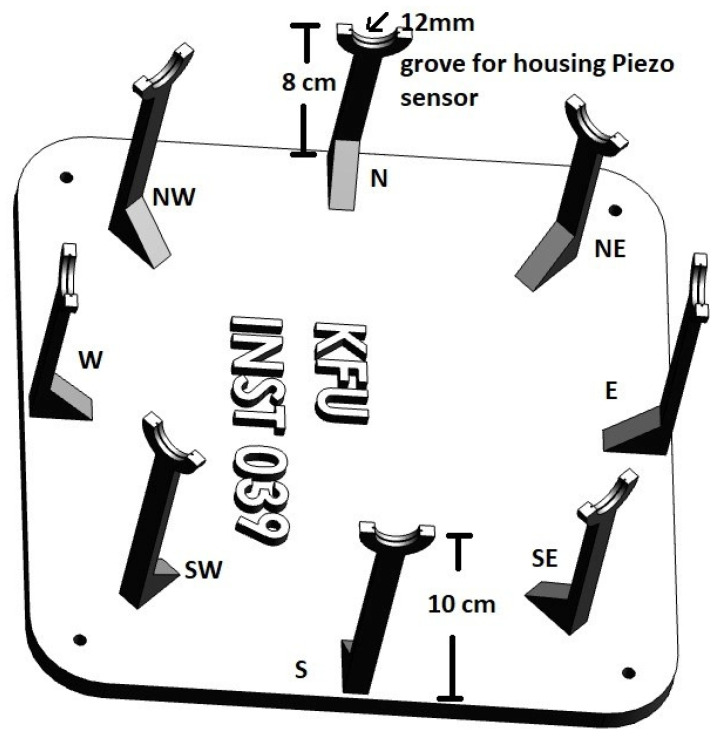
Topview of 3D Printed Prototype for WD Measurement.

**Figure 2 sensors-23-05569-f002:**
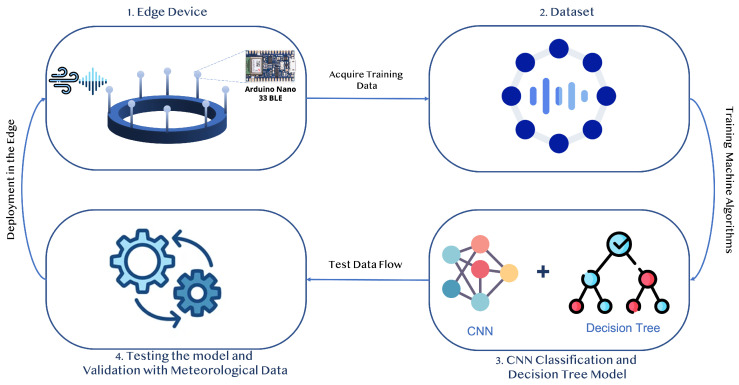
Architecture of Proposed Work.

**Figure 3 sensors-23-05569-f003:**
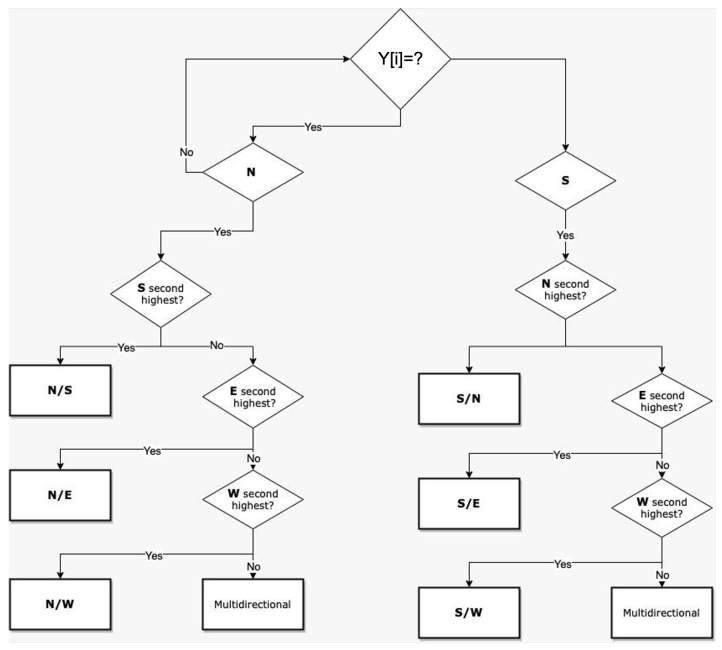
DT Flow Diagram for WD Measurement.

**Figure 4 sensors-23-05569-f004:**
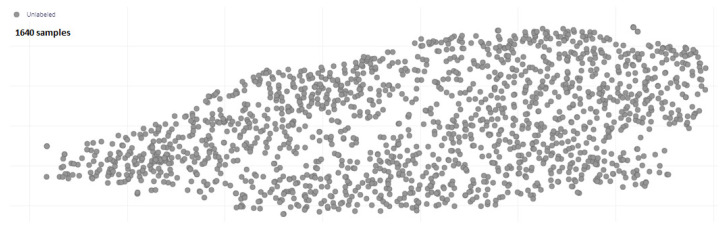
Unlabeled Dataset.

**Figure 5 sensors-23-05569-f005:**
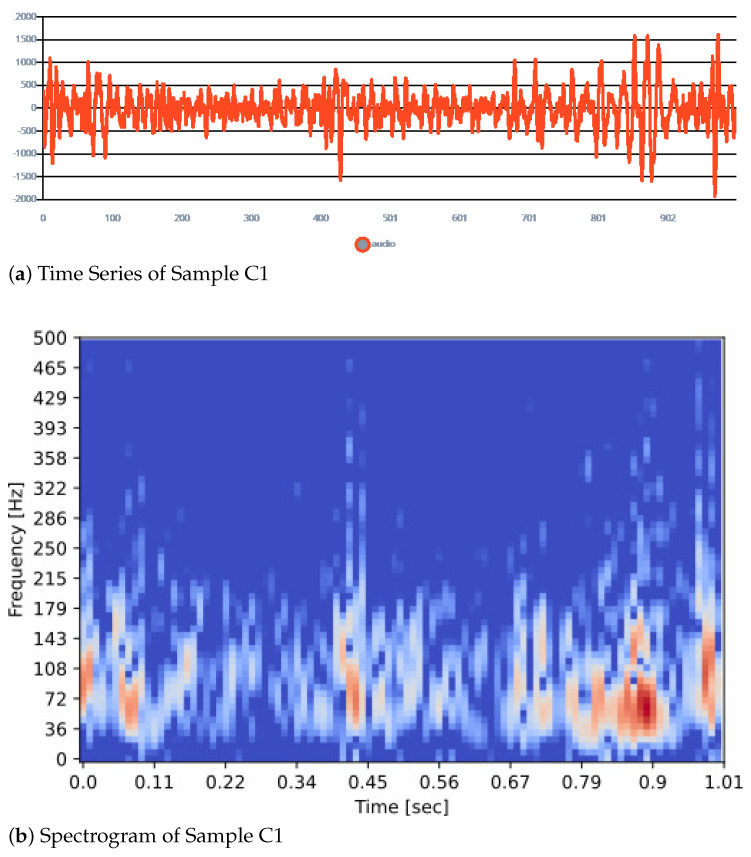
Sample of Subset of Time Series and Spectrogram of Sample Class 1 Used for Feature Extraction.

**Figure 6 sensors-23-05569-f006:**
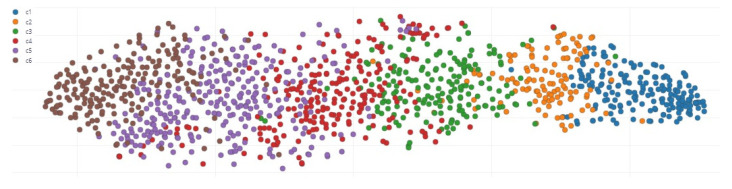
The labeled Dataset for Training the Model in Edge Impulse.

**Figure 7 sensors-23-05569-f007:**
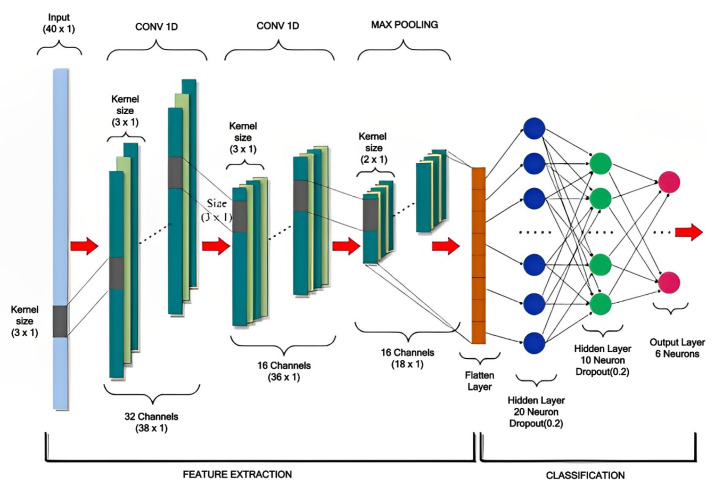
CNN Model Architecture.

**Figure 8 sensors-23-05569-f008:**
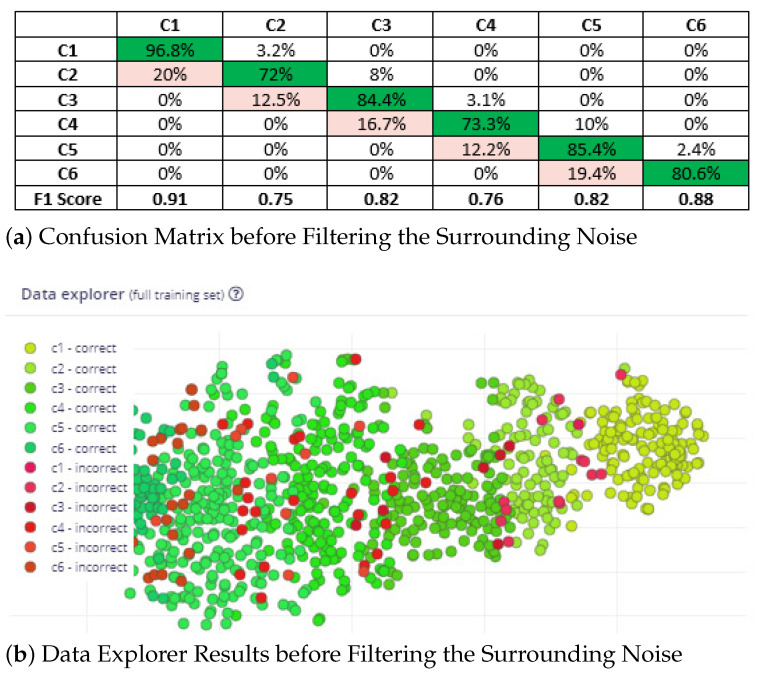
Confusion Matrix and Data Explorer Results Before Filtering the Surrounding Noise.

**Figure 9 sensors-23-05569-f009:**
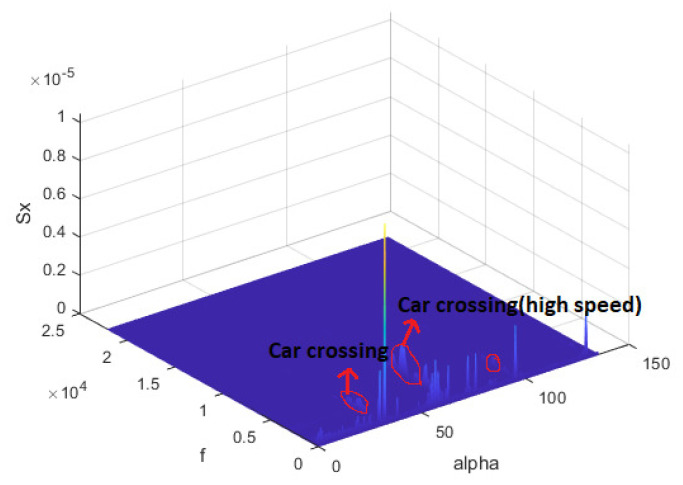
Spectrum plot for the data recorded inside the campus.

**Figure 10 sensors-23-05569-f010:**
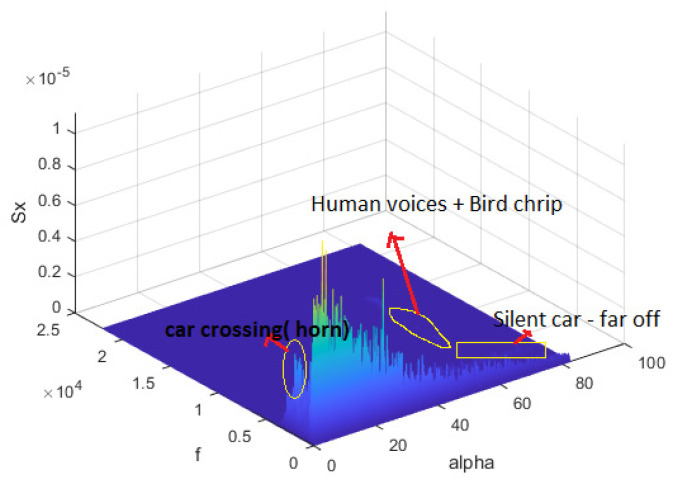
Spectrum plot for the data recorded outside the campus.

**Figure 11 sensors-23-05569-f011:**
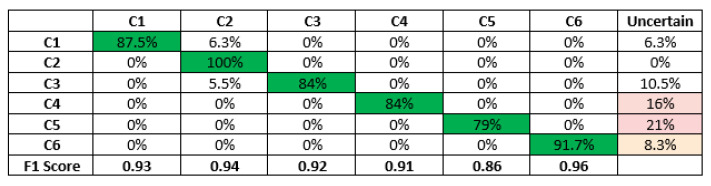
The accuracy and F1 score after filtering the surrounding noise.

**Figure 12 sensors-23-05569-f012:**
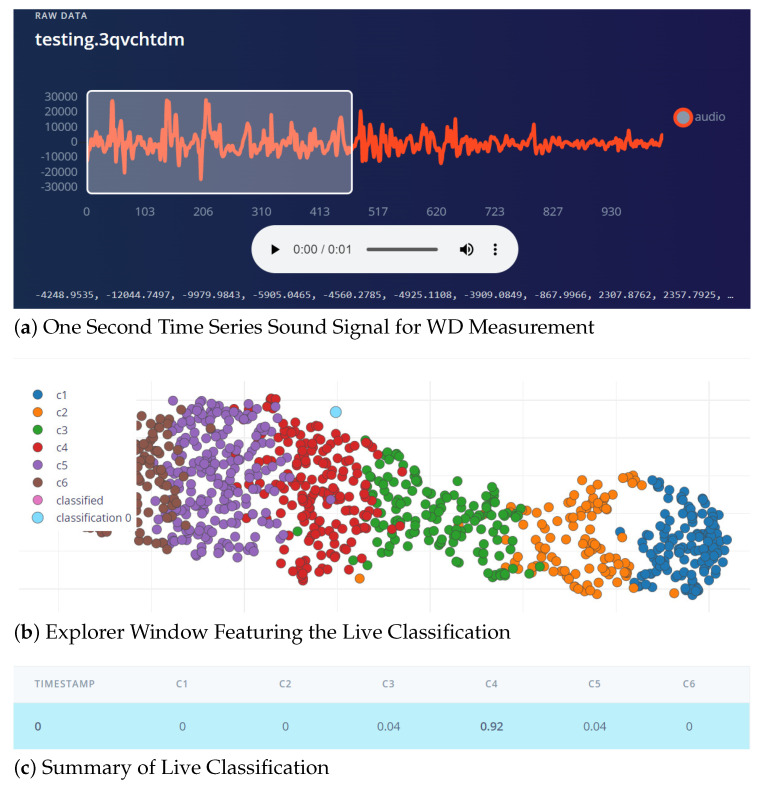
Screenshot of the Live Classification for WV Measurement.

**Figure 13 sensors-23-05569-f013:**
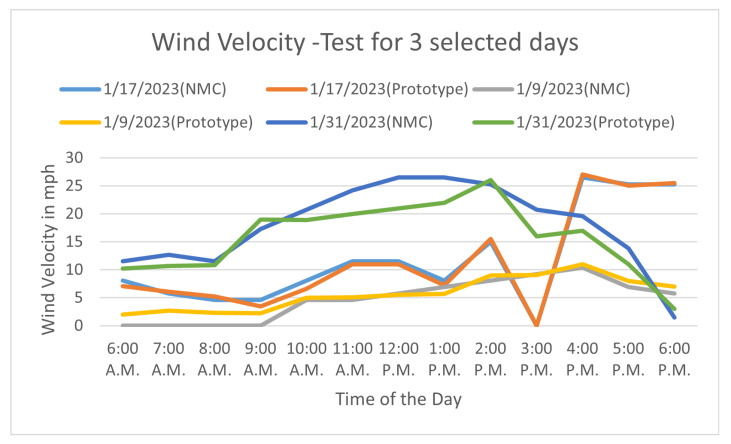
Experimental Results of Wind Velocity.

**Figure 14 sensors-23-05569-f014:**
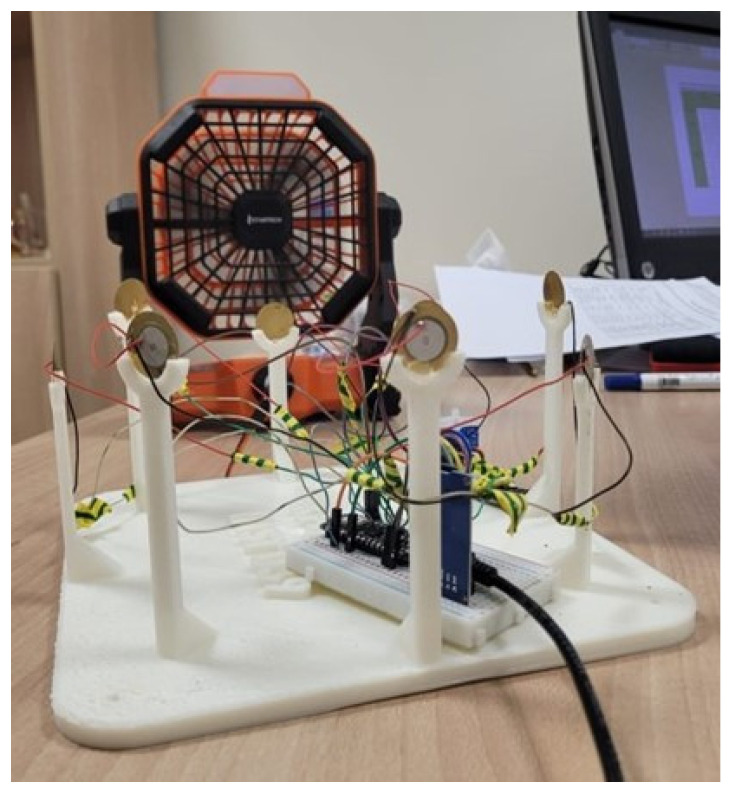
Laboratory setup for the wind direction measurement.

**Figure 15 sensors-23-05569-f015:**
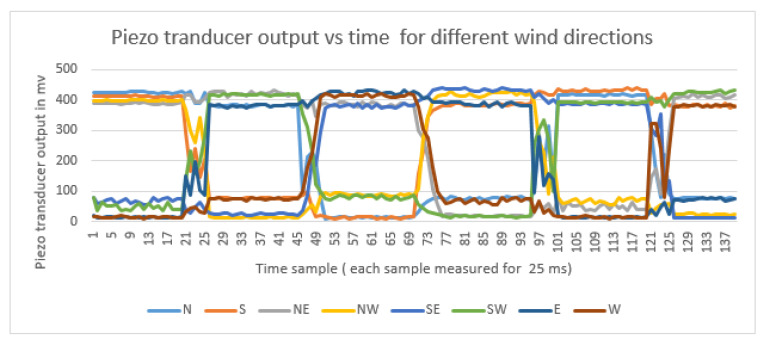
Piezo Transducer Output for Different Wind Directions.

**Figure 16 sensors-23-05569-f016:**
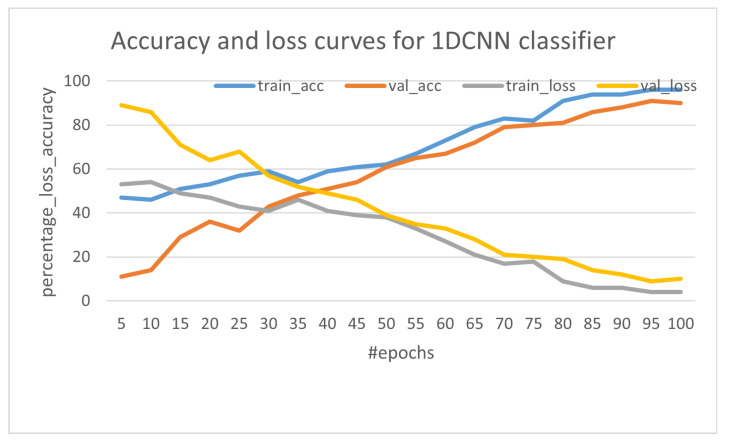
The Accuracy and Loss Curve for Trained Model.

**Table 1 sensors-23-05569-t001:** Comparison of Wireless Technologies for IoT [10,13].

Technology	Standard	Frequency Band	Data Rate	Transmission Range	Energy Consumption	Cost
**Wi-Fi**	IEEE 802.11	2.4–60 GHz	1 Mbps–7 Gbps	20–100 m	Low	High
**ZigBee**	IEEE 802.15.4	2.4 GHz	20–250 Kbps	10–20 m	High	Low
**Bluetooth**	IEEE 802.15.1	2.4 GHz	24 Mbps	8–10 m	High	Low
**MQTT**	OASIS	2.4 GHz	259 Kbps	-	High	Low
**Cellular**	2G/3G/4G/5G	1.8–30 GHz	200 Kbps–20 Gbps	Covered cellular area	Medium	Medium
**LoRaWAN**	LoRa R1	868/900 MHz	0.3–50 Kbps	30 km	Very High	High
**SigFox**	SigFox	200 KHz	100–600 bps	30–50 km	Very High	Low
**WiMAX**	IEEE 802.16	2–66 GHz	1 Mbps–1 Gbps	<50 km	Medium	High

**Table 2 sensors-23-05569-t002:** Wind Velocities Based on BWFS [35].

Wind Force/Class	Description	Wind Speed		
		km/h	Mph	Knots
**0**	Calm	≤1	≤1	≤1
**1**	Light Air	1–5	1–3	1–3
**2**	Light Breeze	6–11	4–7	4–6
**3**	Gentle Breeze	12–19	8–12	7–10
**4**	Moderate Breeze	20–28	13–18	11–16
**5**	Fresh Breeze	29–38	19–24	17–21
**6**	Strong Breeze	39–49	25–31	22–27
**7**	Near Gale	50–61	32–38	28–33
**8**	Gale	62–74	39–46	34–40
**9**	Strong Gale	75–88	47–54	41–47
**10**	Storm	89–102	55–63	48–55
**11**	Violent Storm	103–117	64–72	56–63
**12**	Hurricane	118+	73+	64+

**Table 3 sensors-23-05569-t003:** Model Parameter for 1D-CNN Block.

Parameters	Specifications					
**Model** **Type**	Sequential					
**Input Layer**	3960 feature inputs reshaped to 40 columns using reshape layer					
**First Level 1D-CNN layer**	16 neurons/filters, Kernel size: 3, layer 1Dimensional					
**Second Level 1D-CNN layer**	8 neurons/filters, Kernel size: 3, layer (1Dimensional)					
**1D Pooling(Max)**	Stride = 1, Pool size = 1, Padding = same					
**Activation Function for All Layers**	ReLu					
**Third Level 1D-CNN Layer**	8 neurons/filters, Kernel size: 3, layer (1Dimensional)					
**1D Pooling(Max)**	Stride = 2, Pool size = 2, Padding = same					
**Layer**	Flatten layer					
**Hidden Dense Layer**	16 neurons with drop out rate = 0.2					
**Hidden Dense Layer 2**	8 neurons with drop out rate = 0.2					
**Output Layer**	6 neurons with Softmax activation function					
**Batch Size**	32					
**Epochs**	100					
**Optimizer**	Adam					
**Loss Function**	Categorical Cross entropy					
**Number of Training Cycles**	100					
**Dataset–Training (80%)**	C1 (1–3 mph): 80	C2 (4–7 mph): 70	C3 (8–12): 70	C4 (13–18 mph): 80	C5 (19–24 mph): 90	C6 (25–31 mph): 80
**Dataset–Testing & Validation (20%)**	C1 (1–3 mph): 22	C2 (4–7 mph): 18	C3 (8–12): 18	C4 (13–18 mph): 20	C5 (19–24 mph): 22	C6 (25–31 mph): 20

**Table 4 sensors-23-05569-t004:** Summary of the Classification Results With and Without Filtering.

Class	Metrics before Filtering		Metrics after Filtering	
	Overall Accuracy in %	F1 Score	Overall Accuracy in %	F1 Score
**C1**	94	0.91	96	0.93
**C2**	78	0.75	96	0.93
**C3**	84	0.82	92	0.91
**C4**	80	0.76	92	0.91
**C5**	84	0.82	88	0.86
**C6**	84	0.82	98	0.96

## Data Availability

Data is generated and analyzed during the study.

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
