# Peer review of "Scalable Lightweight IoT-Based Smart Weather Measurement System"

_sensors, 2023, doi:10.3390/s23125569_

Round 1
Reviewer 1 Report
It is an interesting topic. The manuscript organization may be improved before the acceptance.
1. Section 1, 2 and 3 are very closely related to each other according to my opinion, so it may be re-organized.
2. The introduction of the method is not very easy to follow. It seems the authors develop a ML method to invert wind speed and direction through measurements of some variables, for example, the changes of mechanical properties or electronic properties of the sensors. In meteorology, prediction means we use statistical method or numerical model to predict weather from the measurements of current weather condition, but prediction here means we estimate current wind speed and direction from measurements of the sensors. I'm not sure whether my understanding is correct or not.
3. There are so many abbreviations in the manuscript and sometimes they are not explained for the first appearance.
Author Response
Authors’ Response to Reviewer 1 Comments
Authors: Abdullah Albuali, Ramasamy Srinivasagan, Ahmed Aljughaiman, Fatima Mustafa Alderazi
Manuscript ID: sensors-2376678
Title: Scalable Lightweight IoT-based Smart Weather Prediction System
The authors would like to thank the reviewers for the time and effort they spent and providing valuable feedback and are grateful for the insightful comments to improve our paper. We have integrated the suggestions made by the reviewer. We have also reviewed this research again with a professional native language proofreader. Please see below, in blue, for a point-to-point response to the reviewer’s comments and suggestions.
Reviewer Comments:
Point 1: Section 1, 2 and 3 are very closely related to each other according to my opinion, so it may be re-organized.
Response 1:
We re-organized the structure of these sections in such a way that we removed the overlapping between these sections.
Point 2: The introduction of the method is not very easy to follow. It seems the authors develop a ML method to invert wind speed and direction through measurements of some variables, for example, the changes of mechanical properties or electronic properties of the sensors. In meteorology, prediction means we use statistical method or numerical model to predict weather from the measurements of current weather condition, but prediction here means we estimate current wind speed and direction from measurements of the sensors. I'm not sure whether my understanding is correct or not.
Response 2:
We modified through the paper to clarify that this paper measures the wind direction and wind velocity through static sensors that do not have any moving parts as compared to the conventional one. As you mentioned, the prediction in our paper meant estimating the wind velocity and wind direction.
Point 3: There are so many abbreviations in the manuscript and sometimes they are not explained for the first appearance.
Response 3:
We have defined all acronyms on the first appearance of the paper and added the list of abbreviations at the end of the paper.
Reviewer 2 Report
This paper presents the design and implementation of low-cost weather station based on artificial intelligence (AI) algorithm. The work mentioned in the paper is interesting, but the structure of the paper needs to be further modified. For example, there is too little content in Section 5.7 discussion, and Sections 1 to 3 can be considered to be simplified. The overall framework or process of this study should be introduced clearly in the paper first, and it is better to introduce it in combination with frame drawing or flow chart. The title of this paper is weather prediction system, but the realization of this research is only wind speed and direction. Whether the method in this paper is applicable to other meteorological parameters is suggested to add discussion section.
L268:L et al. [5] ,Please check whether L is correct?
L426:Whether it is the right title. Machine learning includes deep learning, which has been mentioned in 3.1. In addition, 3.1 to 3.4 section of the title are relatively short, not conducive to readers grasp the structure of the article, it is suggested that the title description is clear.
L480:The equipment mentioned by the author in this study is low-cost, low-power, reliable, accurate, and easy to install smart weather station. Where can these characteristics be reflected?
Author Response
Authors’ Response to Reviewer 2 Comments
Authors: Abdullah Albuali, Ramasamy Srinivasagan, Ahmed Aljughaiman, Fatima Mustafa Alderazi
Manuscript ID: sensors-2376678
Title: Scalable Lightweight IoT-based Smart Weather Prediction System
The authors would like to thank the reviewers for the time and effort they spent and providing valuable feedback and are grateful for the insightful comments to improve our paper. We have integrated the suggestions made by the reviewer. We have also reviewed this research again with a professional native language proofreader. Please see below, in blue, for a point-to-point response to the reviewer’s comments and suggestions.
Reviewer Comments:
Point 1: This paper presents the design and implementation of low-cost weather station based on artificial intelligence (AI) algorithm. The work mentioned in the paper is interesting, but the structure of the paper needs to be further modified. For example, there is too little content in Section 5.7 discussion, and Sections 1 to 3 can be considered to be simplified.
Response 1:
We restructured the paper and added more contents to Section 5.7 and shortened sections 1 to 3.
Point 2: The overall framework or process of this study should be introduced clearly in the paper first, and it is better to introduce it in combination with frame drawing or flow chart.
Response 2:
We introduced the framework of this paper at the end of the introduction section with a frame drawing or flow chart.
Point 3: The title of this paper is weather prediction system, but the realization of this research is only wind speed and direction.
Response 3:
We have changed the title from prediction to measurement system. This paper focuses only on wind direction and wind speed. As a future work, we will measure other parameters to make it as wind weather prediction system.
Point 4: Whether the method in this paper is applicable to other meteorological parameters is suggested to add discussion section.
Response 4:
We added in the discussion section that the result of this paper can be beneficial to other metrological parameters.
Point 5: L268:L et al. [5] ,Please check whether L is correct?
Response 5:
We have checked the last name of the author and it is correct as the letter L appears as the last name of the author.
Point 6: L426:Whether it is the right title. Machine learning includes deep learning, which has been mentioned in 3.1.
Response 6:
We have restructured that part and made deep learning section as part of the machine learning section.
Point 7: In addition, 3.1 to 3.4 section of the title are relatively short, not conducive to readers grasp the structure of the article, it is suggested that the title description is clear.
Response 7:
We have modified the titles of these sections to be more comprehensive about the entire section.
Point 8: L480:The equipment mentioned by the author in this study is low-cost, low-power, reliable, accurate, and easy to install smart weather station. Where can these characteristics be reflected?
Response 8:
We added the characteristics to reflect the benefits of the utilized sensor in the discussion section.
Reviewer 3 Report
The manuscript presents a lightweight smart weather prediction sensor suite. The low-cost system relies on ML algorithms. The wireless communication for the suite is also discussed. The suite was designed to be used in various applications.
Here are some of my comments and concerns:
The introduction and objective/hypothesis sections are well written.
Acronym "BS" is not defined before when it was first used in line 69.
Related work is quite thorough, and I enjoyed going through it; however, I am concerned how it will keep other readers engaged. It is a bit too long.
All figures need better captions. They need to be as descriptive as possible. You can repeat some of the texts.
It is difficult to read most of the texts in figure 2. It needs to be enlarged or shown better in some other way.
Most of the following figures have smaller texts or blurry images. Those need to be fixed as well.
The results are thoroughly discussed. Besides some confusion in figures, everything else looked good to me.
Overall, I do see some merit in the paper. However, I am still worried about the length of the paper. It is too long and, as I mentioned earlier, it might be difficult to keep the readers engaged.
I would also like to see more in the future work section. It can be a section in itself.
Author Response
Authors’ Response to Reviewer 3 Comments
Authors: Abdullah Albuali, Ramasamy Srinivasagan, Ahmed Aljughaiman, Fatima Mustafa Alderazi
Manuscript ID: sensors-2376678
Title: Scalable Lightweight IoT-based Smart Weather Prediction System
The authors would like to thank the reviewers for the time and effort they spent and providing valuable feedback and are grateful for the insightful comments to improve our paper. We have integrated the suggestions made by the reviewer. We have also reviewed this research again with a professional native language proofreader. Please see below, in blue, for a point-to-point response to the reviewer’s comments and suggestions.
Reviewer Comments:
Point 1: Acronym "BS" is not defined before when it was first used in line 69.
Response 1:
We have defined all acronyms on the first appearance on the paper and added the list of abbreviations at the end of the paper.
Point 2: Related work is quite thorough, and I enjoyed going through it; however, I am concerned how it will keep other readers engaged. It is a bit too long.
Response 2:
We shorten the related work by combining some of the related work that relies on the same approach.
Point 3: All figures need better captions. They need to be as descriptive as possible. You can repeat some of the texts.
Response 3:
We modified the captions of the figures to be more descriptive.
Point 4: It is difficult to read most of the texts in Figure 2. It needs to be enlarged or shown better in some other way.
Response 4:
We modified the text font and size to be clear to readers.
Point 5: Most of the following figures have smaller texts or blurry images. Those need to be fixed as well.
Response 5:
We reinserted most of the images to ensure visibility using a high-quality image extension.
Point 6: The results are thoroughly discussed. Besides some confusion in figures, everything else looked good to me.
Response 6:
We reinserted the images with high quality, changed the font text and size, and made sure the captions are self-descriptive.
Point 7: Overall, I do see some merit in the paper. However, I am still worried about the length of the paper. It is too long and, as I mentioned earlier, it might be difficult to keep the readers engaged.
Response 7:
We shortened the related work and removed some of the overlaps that was exist in section 1 to 3.
Point 8: I would also like to see more in the future work section. It can be a section in itself.
Response 8:
We added a future work section and discussed what issues can be resolved.
Reviewer 4 Report
In this paper the authors present a IoT-based weather station (anemometer for determining the direction and velocity of the wind) that relies on Artificial Intelligence to predict wind direction and velocity.
The subject of the article is interesting and worthy of discussion. The structure of the paper is adequate.
However, there are some aspects that should be clarified and/or analyzed with more detail.
(abstract) Weather prediction accuracy results presented in the abstract must include information about the classes it refers to (nowcast, short-term, medium range, and long-term)
The authors must be consistent when describing the magnitudes that the proposed system allows to measure:
- (Subsection 4.2) "The proposed AWS can measure temperature, wind direction, and wind velocity"
- (abstract) "The proposed work measures multiple weather parameters, such as wind direction, wind velocity, temperature, etc.". What other parameters does "etc" represent?
(Subsection 1.1) The authors identify a set of "Objectives and Hypothesis". However, the topics listed do not represent objectives or hypotheses. They are more like characteristics of the developed device.
In section "3. Related Work" the authors should consider including a final analysis where they highlight the opportunity and motivation of the present study when compared with the related works presented. It is also important to highlight the differences and the innovative contributions of the approach presented in relation to related works.
The authors state that the proposed AWS can measure temperature, wind direction, and wind velocity. However, I didn't find the specifications of the sensor to measure the temperature that was implemented in the system and how this information is processed (I may have missed it).
(Subsection 5.7) "The proposed system runs for approximately X hours in multiple locations". What does the X mean?
I didn't quite understand the purpose of section "5.7. Discussion". Shouldn't the discussion be about the results achieved and how they contribute to the proposed initial objectives?
(Conclusion) The authors state that "we used Rasberri Pi and Arduino to conduct this study". Was the Raspberri Pi also used in the implemented solution?
The figures are adequate but some of them (e.g., Figures 11 and 12) need to be replaced with better quality ones. Additionally, the data presented in some Figures are difficult to analyze due to the type of graph chosen, the number of days considered or even due to the colors of the lines (e.g., Figures 15, 16).
Tables are adequate.
Some references need to be revised (e.g., reference 6 and reference 19 have no date, references 39 and 40 seem to be incomplete, …)
There are some typos in the text (e.g., (line 119) “… Table 2.1.2.”, (conclusion) “Rasberri Pi”, …)
Author Response
Authors’ Response to Reviewer 4 Comments
Authors: Abdullah Albuali, Ramasamy Srinivasagan, Ahmed Aljughaiman, Fatima Mustafa Alderazi
Manuscript ID: sensors-2376678
Title: Scalable Lightweight IoT-based Smart Weather Prediction System
The authors would like to thank the reviewers for the time and effort they spent and providing valuable feedback and are grateful for the insightful comments to improve our paper. We have integrated the suggestions made by the reviewer. We have also reviewed this research again with a professional native language proofreader. Please see below, in blue, for a point-to-point response to the reviewer’s comments and suggestions.
Reviewer Comments:
Point 1: The subject of the article is interesting and worthy of discussion. The structure of the paper is adequate. However, there are some aspects that should be clarified and/or analyzed with more detail.
Response 1:
We addressed all the comments listed below.
Point 2: (abstract) Weather prediction accuracy results presented in the abstract must include information about the classes it refers to (nowcast, short-term, medium range, and long-term)
Response 2:
We have added in the abstract section the classes that we used for measurement. This paper does not predict the wind speed and wind direction. Instead, it measures the current metrological parameters and hence falls in the nowcast class.
Point 3: The authors must be consistent when describing the magnitudes that the proposed system allows to measure:
- (Subsection 4.2) "The proposed AWS can measure temperature, wind direction, and wind velocity"
- (abstract) "The proposed work measures multiple weather parameters, such as wind direction, wind velocity, temperature, etc.". What other parameters does "etc" represent?
Response 3:
We unified throughout the paper what our system can measure. Initially, we interested in relying on temperature, barometric pressure, mean sea level, and relative humidity to predict the weather parameters. However, for this work, we focused only on the wind direction and wind velocity.
Point 4: (Subsection 1.1) The authors identify a set of "Objectives and Hypothesis". However, the topics listed do not represent objectives or hypotheses. They are more like characteristics of the developed device.
Response 4:
We removed the hypothesis term. The main objectives are met by the inherent characteristics of the developed sensors.
Point 5: In section "3. Related Work" the authors should consider including a final analysis where they highlight the opportunity and motivation of the present study when compared with the related works presented. It is also important to highlight the differences and the innovative contributions of the approach presented in relation to related works.
Response 5:
At the end of section 3, we have included the opportunity, motivation, and innovative contributions in relation to the related works. We also added a final analysis table that compares the related work section.
Point 6: The authors state that the proposed AWS can measure temperature, wind direction, and wind velocity. However, I didn't find the specifications of the sensor to measure the temperature that was implemented in the system and how this information is processed (I may have missed it).
Response 6:
The tiny ML kit has inbuilt temperature, pressure, humidity, and other sensors. However, this paper focuses only on the wind measurement while the weather prediction is left for future work. To reduce the length of the paper, the readers are advised to refer to the datasheet of the tiny ML kit to know the sensor specification that measures the temperature and other onboard sensors.
Point 7: (Subsection 5.7) "The proposed system runs for approximately X hours in multiple locations". What does the X mean?
Response 7:
We specified how long our system runs for.
Point 8: I didn't quite understand the purpose of the section "5.7. Discussion". Shouldn't the discussion be about the results achieved and how they contribute to the proposed initial objectives?
Response 8:
We added more discussion content based on the achieved results and clarified how the results in line with initial objectives.
Point 9: (Conclusion) The authors state that "we used Rasberri Pi and Arduino to conduct this study". Was the Raspberri Pi also used in the implemented solution?
Response 9:
In order to minimize the power consumption, we only used the Arduino Nano 33 BLE Sense hardware rather than Raspberry Pi. To avoid any confusion, we removed the Raspberry Pi throughout our proposed work.
Point 10: The figures are adequate but some of them (e.g., Figures 11 and 12) need to be replaced with better-quality ones. Additionally, the data presented in some Figures are difficult to analyze due to the type of graph chosen, the number of days considered, or even due to the colors of the lines (e.g., Figures 15, 16).
Response 10:
We reinserted most of the images to ensure a high-quality image. We also made sure that all texts within images are clear. In addition, we considered different types of graphs to make images easier to analyze.
Point 11: Some references need to be revised (e.g., reference 6 and reference 19 have no date, references 39 and 40 seem to be incomplete, …)
Response 11:
We have revised these references and made sure it is complete.
Point 12: There are some typos in the text (e.g., (line 119) “… Table 2.1.2.”, (conclusion) “Rasberri Pi”, …)
Response 12:
We reviewed the article and removed all the typos.
Round 2
Reviewer 4 Report
In this revision, the authors have made an effort to address some of the concerns given to them in the review.
They improved some parts in several sections (e.g., abstract, Introduction, Related work, discussion and conclusion).
Globally, the manuscript shows improvements over the version previously presented. I believe it was improved enough for meeting publication standards.